# Determining the Fracture Process Zone Length and Mode I Stress Intensity Factor in Concrete Structures via Mechanoluminescent Technology

**DOI:** 10.3390/s20051257

**Published:** 2020-02-25

**Authors:** Seong-Kyum Kim, Ho Geun Shin, Suman Timilsina, Ji Sik Kim

**Affiliations:** 1Department of Civil Engineering, Kumoh National Institute of Technology, 61 Daehak-ro, Gumi-si, Gyeongsangbuk-do 39177, Korea; skim@kumoh.ac.kr; 2School of Nano & Advanced Materials Engineering, Kyungpook National University, Kyeongbuk 37224, Korea; ghrms98@gmail.com

**Keywords:** fracture process zone, stress intensity factor, mechanoluminescent (ML) technology, image segmentation, skeletonization, structural health monitoring

## Abstract

The mechanoluminescent (ML) technology that is being developed as a new and substitutive technology for structural health monitoring systems (SHMS) comprises stress/strain sensing micro-/nanoparticles embedded in a suitable binder, digital imaging system, and digital image processing techniques. The potential of ML technology to reveal the fracture process zone (FPZ) that is commonly found in structural materials like concrete and to calculate the stress intensity factor (SIF) of concrete, which are crucial for SHMS, has never been done before. Therefore, the potential of ML technology to measure the length of the FPZ and to calculate the SIF has been demonstrated in this work by considering a single-edge notched bend (SENB) test of the concrete structures. The image segmentation approach based on the histogram of an ML image as well the skeletonization of an ML image have been introduced in this work to facilitate the measurement of the length of ML pattern, crack, and FPZ. The results show ML technology has the potential to determine fracture toughness, to visualize FPZ and cracks, and to measure their lengths in structural material like concrete, which makes it applicable to structural health monitoring systems (SHMS) to characterize the structural integrity of structures.

## 1. Introduction

The phenomenon of light emissions from certain types of materials in response to different types of mechanical forces or deformation is widely known as mechanoluminescence (ML) [1,2]. A wide range of mechanical stimuli such as tension [3], friction [4], fracture [5], compression [6], torsion [7,8], bending [9], vibration [10], impact [11], and shear [12] can induce ML from materials. Depending on the origin of the ML from a material, whether from destruction or simply from deformation, it can be broadly categorized as fractoluminescence, deformation luminescence, and triboluminescence [1]. The generation of fractoluminescence occurs during the breaking of atomic and chemical bonds in a material, which is mostly accompanied by plasma discharge; therefore, it is destructive and nonreproducible in nature. On the other hand, deformation luminescence occurs without fracturing or damaging the material; therefore, it is reproducible over a number of repetitive mechanical actions. Furthermore, deformation luminescence can be subcategorized into elasticoluminescence and plasticoluminescence depending on the emission of photons in elastic deformation and plastic deformation of the material, respectively. Since the emission of photons in elasticoluminescence has a linear relationship with applied stress or strain, it is commonly applied in stress- or strain-sensing devices. Triboluminescence, which is the third category of ML, is produced either by the contact or separation of two specific types of materials, but it has little potential for use in stress- or strain-sensing devices owing to the fact that there is no linear relationship between the applied stress amplitude and the emitted photons.

Materials that possess elasticoluminescence are widely used in the production of ML paint and ML sheets for detecting cracks and assessing the crack-tip stress field at the crack-tip vicinity lying either on the surface or on the body of a structure [12,13,14,15,16]. Detecting cracks, tracing the propagating path, and assessing the crack-tip stress field are important steps to be undertaken in preventing the catastrophic failure of a structure, because every material is susceptible to fracture due to repetitive and extreme mechanical forces and to the chemical and thermal environments a material faces during its lifetime. To prevent the catastrophic failure of a structure, several experimental methods have been proposed, and these are either one-dimensional or two-dimensional measurement techniques. The latter type, which includes electronic speckle interferometry [17], stereography [17], radiography [18], thermography [19], and guided wave imaging [20], is considered important as it enables direct visualization of cracks; however, all of them require both highly sophisticated measurement devices and postanalysis processes that are time-consuming and complicated. ML technology, on the other hand, comprises stress-/strain-sensing ML paint or ML sheets accompanied by an imaging system and digital image processing technique with the potential to reveal surface cracks, body cracks, and fatigue-induced microcracks [21] in addition to an ability to elucidate fracture-induced phenomena such as elastic crack-tip stress fields [12,13,14] and plastic crack-tip stress fields [12], and to measure stress [22] in both simple and complex structures. These advantages are why ML technology has drawn significant attention for its simplicity and cost-effectiveness as a whole-field and noncontact measurement system. Since the discovery of elasticoluminescence, ML technology has been used over the past two decades to study the fracture mechanisms in a wide range of ceramic materials such as Al_2_O_3_, Si_3_N_4_, and zirconia-based ceramics, in metallic materials such as aluminum, and in concrete [1,2,23]. Furthermore, ML has been used in the detection of multiple cracks in mega structures such as bridges [24], buildings, and pipes [25], which ensures the utility of this technology in structural health monitoring systems [26,27,28]. Moreover, ML paint has become an integral part of ML technology because of weatherproof characteristics that promote excellent mechanical and optical durability, which has further established its applicability in megastructures where the aforementioned features are deemed very essential. 

As previously mentioned, ML technology has been applied to a wide range of structural materials to explain the crack-initiating mechanisms, crack-driving mechanisms, and the crack-tip stress fields both in elastic and plastic regions, as well as in the bridging field; however, the potential use of ML technology to visualize and explain the fracture process zones (FPZs) that are commonly found in concrete structures has never been done. In fact, the existence of FPZs in concrete structures has prevented the ability of linear elastic fracture mechanics (LEFM) to accurately address the fracture mechanisms of concrete. As a consequence, several theories concerning the effect of FPZs have been proposed [29] to elucidate the fracture mechanics of concrete. To be effective, however, these theories must be verified and upgraded, which require an accurate FPZ length [30,31,32,33]. ML technology could be used to simply fetch the FPZ dimensions. Therefore, in this work, to understand the development and advancement of FPZs via ML technology, two different types of concrete samples were considered with and without the incorporation of aggregate. To identify an FPZ and measure its length and image segmentation based on an image histogram has been introduced. Furthermore, image skeletonization was considered in order to improve the technical proficiency of measuring an FPZ and the crack length from an ML image. Finally, the crack length was measured using both a skeletonized ML image after the subtraction of FPZ and crack mouth opening displacement (CMOD: which is an indirect method to measure crack length) in order to determine the Mode I stress intensity factor (SIF) for the samples considered. The Mode I SIF is the most often used engineering design parameter in fracture mechanics which reveal the fracture tolerance of materials used in bridges, buildings, aircraft, and so forth. Both the FPZ and the SIF are important parameters that are needed to diagnose the integrity of structural materials.

## 2. Materials and Methods

In order to visualize and measure the FPZ length, two samples of concrete were formed with different levels of fracture toughness. Concrete from mortar only (MC) and concrete that included aggregate (AC) each were formed following normal laboratory procedure. To form the AC, cement, sand, and aggregate (diameter ~10 mm) were homogeneously combined followed by the addition of water with constant mixing for five more minutes to form a paste that was then poured into a 400 mm × 50 mm × 50 mm (Length × Breadth × Thickness) beam mold. To create a notch on the specimen, a 50 mm × 50 mm × 1 mm (Length × Breadth × Thickness) steel plate was inserted on the casting side. The MC was prepared following the same procedure without the addition of aggregate. Thereafter, the samples were allowed to stand under lab conditions (25 °C and 50% relative humidity: RH) for 12 h to allow solidification. Subsequently, the solidified samples with the dimensions shown in Figure 1a were demolded and further cured for 24 h in a chamber that maintained high levels of temperature and humidity (100 °C and 100% RH) to enhance the strength. The details regarding the concrete components are highlighted in Table 1.

Stress-sensing elasticoluminescent SrAl_2_O_4_: Eu, Dy (SAOED) microparticles with an average diameter of 20 µm were purchased from NEMOTO & CO., Japan. ML paint was prepared by homogeneously mixing SAOED and optically transparent epoxy resin in a weight ratio of 3:7 at 25 °C in a centrifugal planetary mixer. The homogeneously mixed epoxy-SAOED composite was degassed for 10 min before it was applied to the samples. After applying the paint in front of the notch on the concrete sample, as illustrated in Figure 1b, the sample was allowed to stand for 24 h to allow the paint to solidify. It should be noted that ML paint thickness should be very small as compared to the thickness of concrete; otherwise, the greater thickness of paint could affect the deforming characteristics of concrete. Furthermore, the paint must be uniformly applied to avoid the effect of paint thickness on the emission. Thicker paint emits brighter ML than thinner paint; therefore, uniformity in the paint layer is important to minimize thickness effects while analyzing structural deformation.

For the SENB test, the sample was laid on a loading stage that was specially designed for SENB testing, and a CMOD gauge was attached at its notch, as shown in Figure 1b. The paint on the sample was exposed to UV light for 5 min and then aged in the dark for 2 min to allow the long phosphorescence to relax to a reasonable level. After this, a 2000 N/min load was applied until the concrete fractured. During loading, the painted surface was photographed on a macro scale at a frame speed of 250 frames/s using a high-speed imaging system. The data recorded from a load cell and CMOD were synchronized with each frame using a multichannel data link (MCDL). The experimental setup is schematically illustrated in Figure 1c. 

## 3. Results and Discussion

The load-CMOD response curves of AC and MC illustrating their levels of fracture strength under SENB testing are illustrated in Figure 2. The load first increased rapidly to the peak level and then dropped with the propagation of cracks in both samples. It is well understood that cracks in concrete tend to move along a path either where the interfacial zone is weakest or through any existing pores. If a crack encounters aggregate particles, it is forced either to move through the stronger aggregate or to deflect and propagate around the weaker aggregate–mortar interface. In both situations, a large amount of energy is absorbed; consequently, the AC showed fracture strength that was higher than that of the MC. 

Figure 3 shows representative sequential ML images of crack initiation and propagation in two samples. The images at 0 N showed faint emissions, which was attributed to the long phosphorescence of SAOED. The advancing ML pattern in AC and MC was vivid at peak loads of 1.46 and 1.04 kN, respectively, as well as for onward loads. Even before the peak load, a faint ML pattern that developed at the notch tip was apparent in both specimens at load levels of 0.2 and 0.1 kN, respectively, in both AC and MC. The ML pattern in the AC was much higher than that in the MC, which resulted from the difference in fracture strength. The length of the ML pattern was then compared with the crack length obtained from the CMOD gauge, as illustrated in Figure 4. Figure 4a compares the ML pattern with the crack length calculated from the CMOD for the AC while Figure 4b shows that of the MC. In both samples, the overall ML pattern length was higher than the crack length that was calculated based on the CMOD gauge for a given load. This difference clearly shows that the ML pattern was not generated solely from the crack but also from the microcracking region, also known as the FPZ, ahead of the traction-free crack tip, which the CMOD does not measure.

The CMOD was measured using TML Dynamic Strainmeters with sampling frequency of 5 kHz. The CMOD data from TML Dynamic Strainmeters was further synchronized with camera images and applied load using MCDL with 250 sample/s. Therefore, each image was taken for every 0.004 s and the total duration of whole experiment shown in Figure 2 was 2.844 s. Since the concrete is brittle in nature, the crack speed under 2000 N/min loading rate was very high resulting in very short duration for crack propagation (around 0.064 s). As a result, out of 500 synchronized data points, around 16 images only depicted crack propagation, which were considered for the assessment of the FPZ. 

Understanding the FPZ is essential due to the invalid nature of linear elastic fracture mechanics (LEFM), inaccuracies in the predictions of failure for concrete structures, and the fracture energy dependency on FPZ size. Therefore, an accurate size for FPZ is helpful when using the finite element method (FEM) to simulate the crack propagation path and for the development of new theoretical models [29,30,31,32,33]. Techniques that involve X-ray, acoustic, interferometry, and optical microscopic analyses have been employed to elucidate the FPZ phenomenon. The aforementioned methods, however, are time-consuming and inaccurate, and they require testing instruments that generate mechanical vibration, require advanced sample preparation techniques, and are prohibitively expensive. To overcome these disadvantages, digital image correlation (DIC) techniques have been employed by some researchers where digital images obtained from high-speed imaging systems are analyzed to measure the FPZ based on strain or displacement fields [33]. Both DIC and ML technologies use an imaging system to record the fracture process and the postprocessing of digital images to determine various fracture parameters, which further promotes the use of ML images to assess the FPZ.

Before the estimation of FPZ length, it is first necessary to understand the stress-field distribution around the crack-tip vicinity in concrete. The stress profile in the front of the crack tip along the crack-propagating axis in concrete is shown schematically in Figure 5a, where the peak stress lies in the FPZ rather than at the crack tip. A shift in the peak stress away from the crack tip has been observed in elastic–plastic material via ML technology. The length of the shift in the peak stress determined using ML technology approximated the length obtained using Irwin’s method for finding a virtual crack tip based on the stress intensity factor of the material [12]. Unlike isotropic elastic–plastic material where the stress field is concentrated at a particular point, the inhomogeneity characteristics of concrete and the distributions of several microcracks, as shown in schematic Figure 5a, show that the stress field is highly distributed around microcracks, which makes it difficult to pinpoint the peak stress point at the FPZ when using ML technology. However, the FPZ stress field should trigger an ML pattern, as the lighter green color reveals in Figure 5b. In fact, the stress field in an FPZ is transferred to stress-sensing, microsized SAOED particles through the binder, which then triggers an emission from the SAOED particles that results in the formation of an ML pattern. The photon emission from SAOED can be described using a trap-controlled mechanism [1]. An electron from Eu^2+^ is released to conduction band during UV irradiation and gets trapped by oxygen vacancy. Thereafter, the trapped electron is released by mechanical stimuli and is recombined with the oxidized Eu^3+^, which results in the emission of photons, consequently creating ML patterns [1,2,23]. The larger length of ML pattern than the length of crack measured using CMOD clearly indicates an emission from the microcracking zone as clarified in Figure 5; however, it is difficult to discern the difference between an emission from the propagated crack and one from the FPZ. In principle, the emission that originates from the crack propagation should be higher than the emission from a microcracking zone since stress concentration after the coalescence of microcracks will be localized and consequently results in a higher level of emission. However, due to dispersion of the stress fields in the FPZ, emissions from the FPZ should be fainter compared with those from the main crack, as illustrated in Figure 5b. To distinguish between an emission from a main crack and that from a microcracking zone, it is necessary to adopt a digital image analysis technique; therefore, image segmentation based on an image histogram was adopted to distinguish the emission patterns in this work.

An image histogram is a graph of the pixel intensity (on the *x*-axis) versus the number of pixels (on the *y*-axis) [34,35]. The histogram of an ML image (Figure 6a) is depicted in Figure 6b, where the *x*-axis represents the gray level depicting the intensity value and the *y*-axis represents the total number of pixels. From a histogram, an ML image can be broadly categorized into three different parts: the First Zone (0–51), the Second Zone (52–178), and the Third Zone (179–256). The variation in pixel intensity is also depicted by the small square-shaped image parts in Figure 6b. The First Zone is free from any stress and any emission from this zone originates from the enduring phosphorescence of the ML paint. The Second Zone has an intensity gradient that is distinctive from those of both the First and Third Zones, and is believed to originate from the microcracking zone around the main crack. A wide range of pixel values in this zone suggest a nonuniform stress distribution owing to the fact that there typically are different stress fields in the vicinity of a microcrack and between the microcracks in a region. Moreover, the densities of microcracks vary as they extend away from the traction-free main crack, which also results in a degrading of the intensity away from the traction-free main crack. The Third Zone shows the emission from the main crack, where the stress localization should result in the brightest zone in the ML image. Based on this histogram, the image is segmented into three respective zones as illustrated by the different colors in Figure 6c [36,37]. The First Zone is highlighted in green, the Second Zone is highlighted in blue, and the Third Zone is highlighted in red. The Second and Third Zones are also shown separately in Figure 6d,e, respectively. In the segmented Figure 6d,e, the length of the FPZ can be calculated by measuring the gap length.

In order to measure the crack length based on the ML image, it is important to develop new methods to reduce time and increase accuracy, which is impossible with a manual measurement using the ruler in Photoshop. For example, in an isotropic composite material where a clear localized crack-tip stress field distribution is observable from the ML, the crack tip is easily determined because the maximum ML emission lies at the crack tip, which is the site of the maximum concentration of stress. Therefore, knowing the crack-tip position simplifies the measurement of the crack length and allows the instantaneous calculation of fracture parameters such as the J integral and the SIF when using MATLAB software [12]. The nonhomogeneous nature of a concrete surface, however, can result in the aforementioned nonconcentrated and spatially distributed stress fields, which can make it impossible to determine a crack tip akin to the experience of working with isotropic homogeneous materials. Therefore, a new method is essential. In the present work, crack and FPZ measurements using skeleton images of the ML were introduced in order to facilitate their automated measurement.

Skeletonization provides an effective and compact representation of an object by reducing its dimensionality to a “medial axis” or “skeleton” while preserving the topologic and geometric properties of the object. Since Blum introduced the concept of skeletonization [38], it has been applied to numerous forms of image processing and computer-vision applications such as object description, matching, classification, and so forth. For skeletonization, several different types of computational approaches exist, and their fundamental principles are mentioned in the literature [39,40]. In fact, many researchers have undertaken skeletonization based on continuous approaches, while others have preferred purely digital methods for skeletonization. In this work, skeletonization was undertaken using the MATLAB library code, and the results are illustrated in Figure 7.

The first step involves changing ML images from a high-speed imaging system into binary images, after which the skeleton images are obtained using the MATLAB library code. A representative binary image and its skeleton image are depicted in Figure 7a,b, respectively. Then, segmented images showing the crack are converted into binary images and subsequently into their skeleton images. A representative binary image and its skeleton image are depicted in Figure 7c,d, respectively. The length of the ML pattern in Figure 7a, which consists of both an actual crack and its FPZ, is measured simply from the ends of the skeleton coordinates along the *y*-axis, Y_1_-Y_o_.The crack length in Figure 7c was calculated in a similar fashion, y_1_-y_o_. Thereafter, the length of the FPZ was determined by subtracting y_1_ from Y_1_ (i.e., Y_1_-y_1_). Therefore, measuring the crack length and FPZ using skeletonized images is much easier; moreover, the accuracy is increased since a length equal to a single pixel can be measured accurately. In fact, measuring the ML pattern with spatial distribution in the *xy* plane is a tough job for the naked eye using Photoshop’s ruler, because it is not easy to determine the ends of the ML pattern, which results in errors as well as in a great requirement of time and effort. These facts are why approaches of image skeletonization have been undertaken by other researchers to measure the crack length and visualize the crack path [41]; however, the FPZ length has never been measured, particularly in the ML field.

Following the aforementioned procedure, crack and FPZ lengths were measured for both samples in the present study, as illustrated in Figure 7e. The illustration shows that with the advancement of the crack, the FPZ increased first and then dropped in both samples. The FPZ length that was measured using the DIC technique showed the same trend, but it monotonically increased and dropped without fluctuating as in this method [33]. The fluctuating FPZ length with crack advancement likely originated from the nonhomogeneous nature of concrete in conjunction with the high sensitivity of ML paint. In fact, a highly sensitive type of paint or patch is desirable for the sake of understanding the crack vicinity fracture phenomenon of a structure under a static or dynamic loading environment. For example, paint revealed the FPZ before the advancement of the crack, as illustrated in Figure 3. At this stage, the CMOD showed a crack length almost equal to the initial notch length, but paint revealed the development of an FPZ. This information is essential for structural health monitoring and in fact more advanced stress-sensing paint is desirable so that stress distribution at the crack vicinity can be visualized even without the early development of an FPZ. The realization of such paint is possible by using nano-/microsized stress-sensing particles with the ability to respond to conditions of low stress; unfortunately, one of the best stress-sensing particles used in this type of work emits photons after a threshold stress level of 5 MPa [13].

After the determination of an FPZ, it is worthwhile to determine the stress intensity factor of the samples based on the crack length using an indirect method as well as the crack length using the image after the exclusion of the FPZ. The SIF was calculated using Equation (1), which is a commonly used SENB test for concrete [42].
(1)$$SIF=K_{I}=(\frac{3PS}{2W^{2}B})\sqrt{\pi a}.f(\frac{a}{W})$$
$$\begin{array}{l} \mathrm{where}\\ P\;=\;\mathrm{applied}\;\mathrm{load}\\ S\;=\;\mathrm{span}\\ W\;=\;\mathrm{specimen}\;\mathrm{depth}\\ B\;=\;\mathrm{specimen}\;\mathrm{thickness}\\ a\;=\;\mathrm{crack}\;\mathrm{length} \end{array}$$
(2)$$f\left(\frac{a}{W}\right)=\frac{1.99-\left(\frac{a}{W}\right)\left(1-\frac{a}{W}\right)\left[2.15-3.93\frac{a}{W}+2.70{\left(\frac{a}{W}\right)}^{2}\right]}{\sqrt{\pi }\left(1+2\frac{a}{W}\right){\left(1-\frac{a}{W}\right)}^{\frac{3}{2}}}=\mathrm{geometric}\;\mathrm{factor}$$

The results are illustrated in Figure 8. Figure 8a compares SIFs obtained from the indirect method and from the ML crack. The agreement between the two methods was as good as a/W= 0.7; however, once the crack propagated further, the SIFs diverged. Unlike in AC, the SIFs in MC showed perfect agreement between the two methods. The divergence of the SIF is directly related to the slight mismatches between the crack lengths obtained from the two different methods. In fact, for a given load and crack length, the SIF increases geometrically with an increase in the a/w, as indicated by Equation (2), and, therefore, despite a slight mismatch in the crack length, the SIF tends to show a gap, and this gap increases with a higher a/W.

This work has clearly shown that ML technology can reveal the development and advancement of the FPZ that is most commonly found in concrete structures as an addendum to the actual measurement of real crack length. Also, the SIF, which is a very important material characteristic, can be calculated using ML technology. Using CMOD can only provide information about crack length, but ML technology can provide full-field information about the crack path, crack tip, multiple cracks, and the distribution of the stress field at the crack-tip vicinity in real time. These features are desired in developing a structural health monitoring system to characterize the structural integrity of large structures such as bridges and buildings [24,43]. ML technology can be a reliable and effective tool for SHMS because it is a full-field and noncontacting measurement system with a short response time that is cost-effective and offers simplicity in application and data processing. Nonetheless, there is always room for enhancing the effectiveness of ML technology which mainly depends on a few improvements: first, the discovery of noble stress/strain-sensing novel particles with very low threshold stress, so that the emission of photons is guaranteed under minimally applied stress; second, the discovery of optimized binders where the stress transferability from binder to stress-sensing particles is high; as a result, the emission of photons is enhanced remarkably. Last, the incorporation of advanced digital image processing techniques is necessary, so that several features of fracture mechanics can be revealed effectively in less time via ML image analysis.

## 4. Conclusions

A new method to measure the FPZ length and calculate the SIF based on ML technology has been demonstrated in this work by considering the SENB testing of two different concrete structures with and without aggregate. The length of the FPZ was calculated by subtracting the visualized crack length from the visualized ML pattern after the ML image segmentation into a main crack and microcrack sections. The image segmentation was accomplished by adopting an image histogram in this work and was a very effective method despite its simplicity. To more simply measure the crack length and the FPZ, a skeletonizing technique was introduced, which was very useful in measuring the length of the patterns in the ML images. The measured FPZ showed a tendency to first increase and then drop, which was a trend reported by other studies. Fluctuation in the increase and decrease in the FPZ length was attributed to the sensitivity of the ML paint as it revealed the inhomogeneous nature of a concrete structure. Furthermore, another important fracture parameter, SIF, was calculated based on the crack length measured using both the ML technology and the indirect CMOD method. Even though the SIFs from both methods showed a perfect match in the sample without aggregate, a slight mismatch in crack length resulted in different values of SIFs for the two different methods in the sample with aggregate. Nonetheless, as a whole-field and noncontact method to determine fracture parameters such as fracture toughness and FPZ length in structural material like concrete, ML technology has enhanced its potential use in structural health monitoring systems (SHMS) to diagnosis the structural integrity of megastructures. 

## Figures and Tables

**Figure 1 sensors-20-01257-f001:**
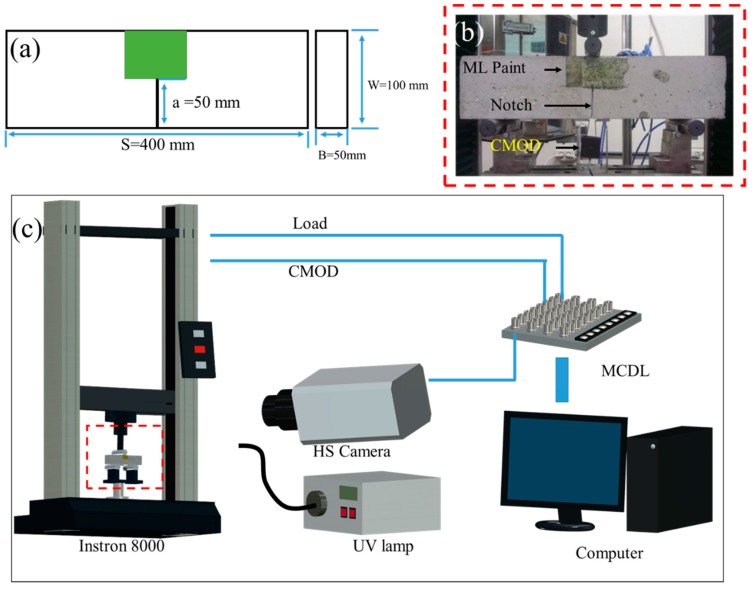
(**a**) Schematic diagram of a single-edge notched bend (SENB) specimen showing its dimensions with ML paint (green patch). (**b**) Real image of MC with ML paint applied to the surface ahead of the notch and a crack mouth opening displacement (CMOD) gauge attached to its notch. (**c**) Schematic diagram of the experimental setup illustrating the Instron 8000, UV lamp, high-speed camera, multichannel data link (MCDL), and computer that were used in the experiment.

**Figure 2 sensors-20-01257-f002:**
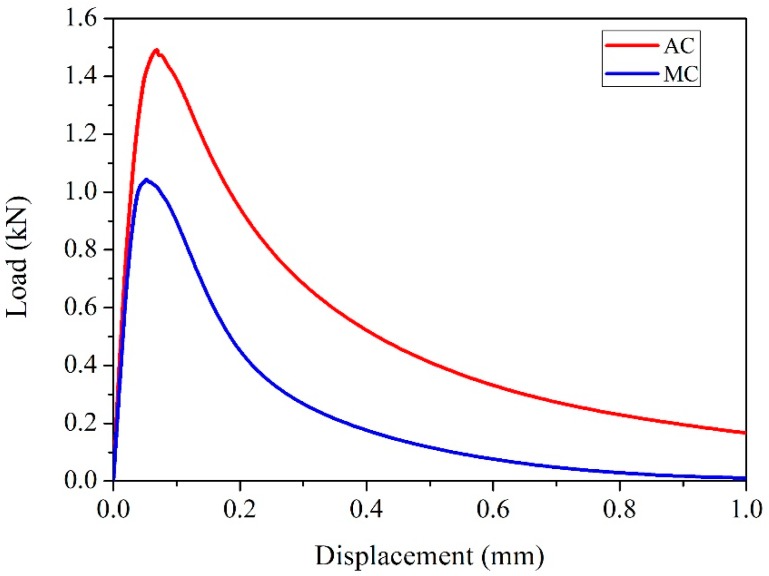
Load-CMOD relation curves of AC and MC illustrating their different levels of fracture strength.

**Figure 3 sensors-20-01257-f003:**
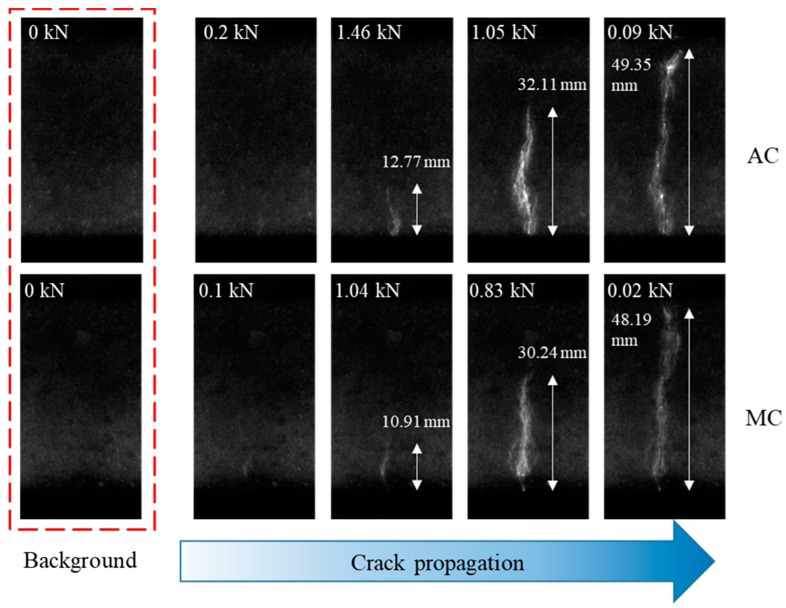
Sequential ML gray images illustrating crack initiation and propagation in AC and MC during SENB testing, which were recorded using a high-speed camera at a frame speed of 250 frames/s.

**Figure 4 sensors-20-01257-f004:**
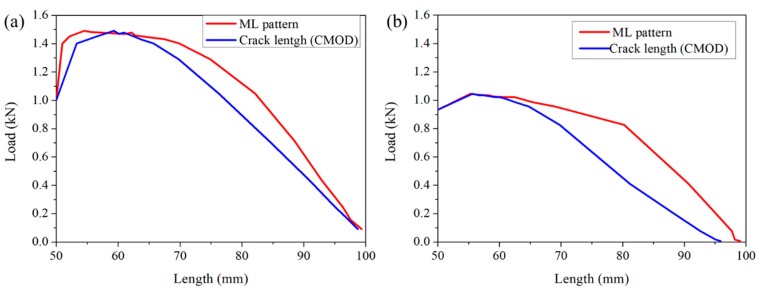
Comparison of the ML pattern length and crack length calculated using the indirect method (CMOD) for AC (**a**) and MC (**b**).

**Figure 5 sensors-20-01257-f005:**
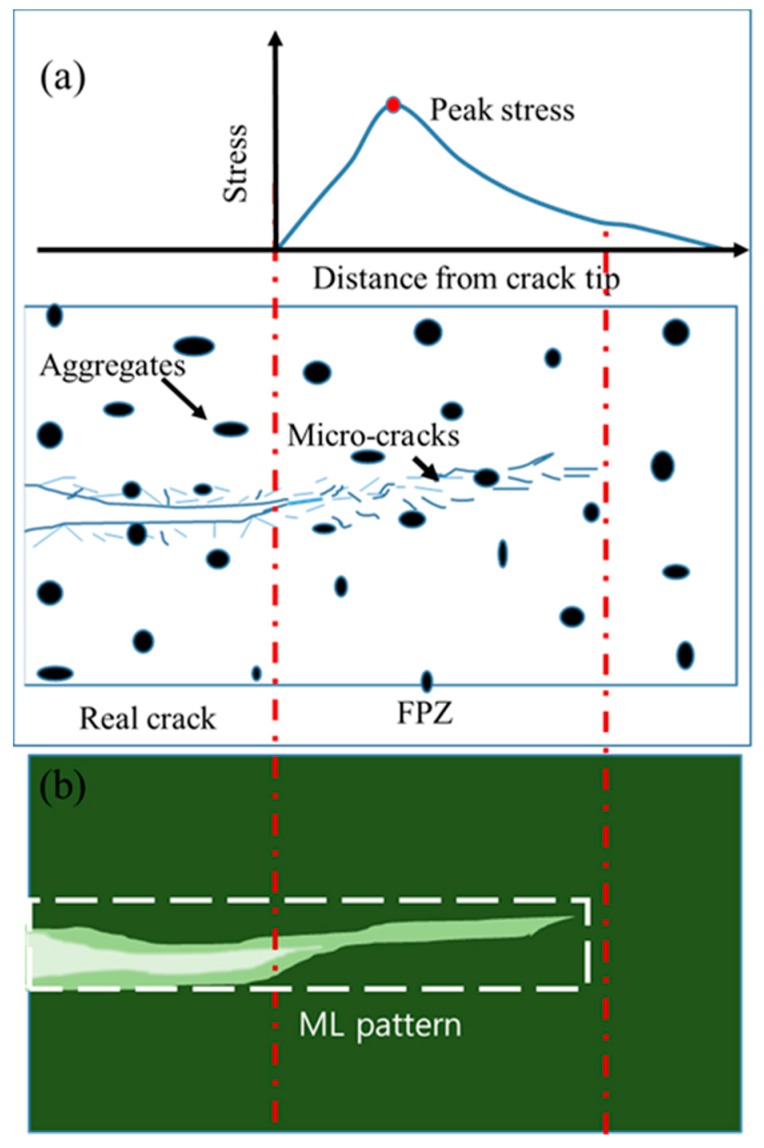
(**a**) Stress profile ahead of a crack tip in concrete material where the FPZ has the maximum stress concentration. (**b**) Schematic diagram showing the ML intensity from ML paint on a concrete sample. Among the three different colors, the dark green illustrates long phosphorescence, the lightest green depicts the emission from a recently propagated crack, and the remaining lighter green is the emission from a microcracking zone.

**Figure 6 sensors-20-01257-f006:**
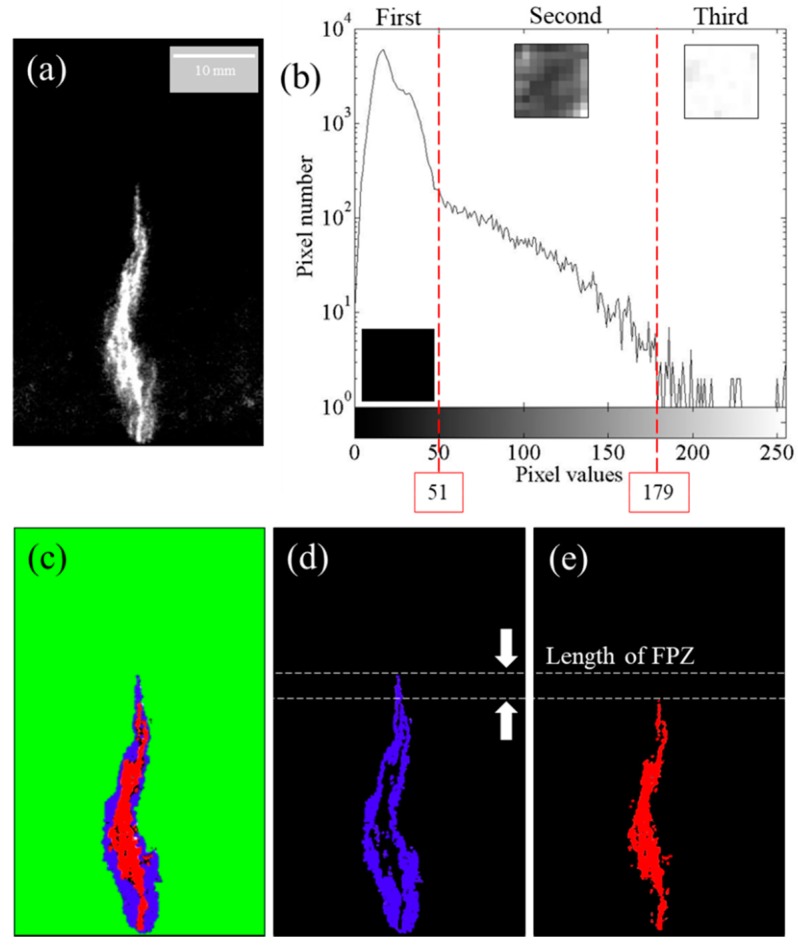
The histogram of (**a**) has been derived using MATLAB software, which is illustrated in (**b**). Based on the histogram, (**a**) has been segmented into three different zones as illustrated in (**c**) by green, blue, and red colors. The Second Zone and the Third Zone are illustrated separately in (**d**,**e**), respectively; meanwhile, the length of the FPZ is illustrated from (**d**,**e**).

**Figure 7 sensors-20-01257-f007:**
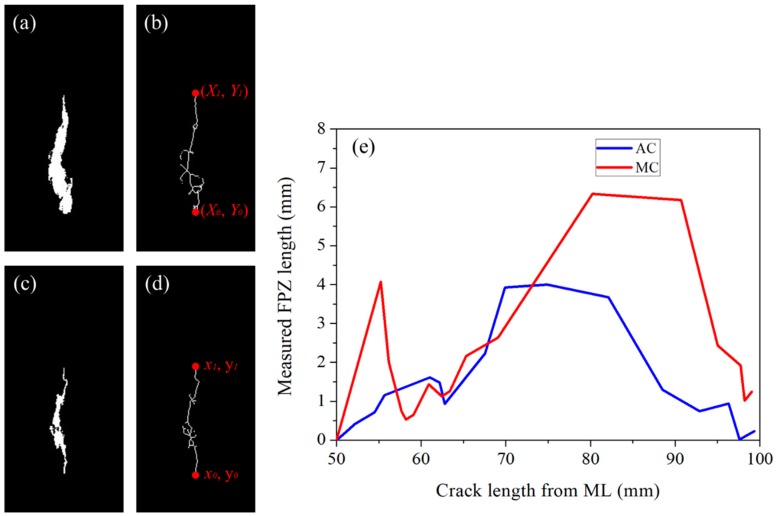
(**a**) Binary ML image of Figure 6a showing the ML pattern. (**b**) Skeleton image of (**a**). (**c**) Binary image of Figure 6e showing the crack after the exclusion of the First Zone and the Second Zone. (**d**) Skeleton image of (**c**). (**e**) Comparison of the FPZs of two different samples with crack lengths obtained from the skeleton images.

**Figure 8 sensors-20-01257-f008:**
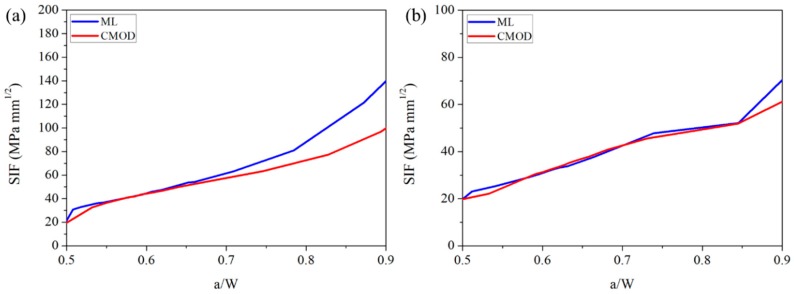
Comparison of SIFs that were obtained from an indirect method (CMOD) and by using ML technology, where (**a**) shows the SIFs of AC and (**b**) shows the SIFs of MC.

**Table 1 sensors-20-01257-t001:** The proportion of materials used in the specimens.

Specimens	$\mathrm{Water}\;\left({\mathrm{kg}/m}^{3}\right)$	$\mathrm{Cement}\;\left({\mathrm{kg}/m}^{3}\right)$	$\mathrm{Sand}\;\left({\mathrm{kg}/m}^{3}\right)$	$\mathrm{Aggregate}\;\left({\mathrm{kg}/m}^{3}\right)$
AC	195.23	356.63	839.68	859.21
MC	195.23	356.63	1069.89	-

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
