# Peer review of "Determining the Fracture Process Zone Length and Mode I Stress Intensity Factor in Concrete Structures via Mechanoluminescent Technology"

_sensors, 2020, doi:10.3390/s20051257_

Round 1

Reviewer 1 Report

In this paper, the authors explain the use of mechanoluminescence to use of ML technology to visualize and explain the fracture-process zones (FPZs) in concrete structures as well as calculate the stress intensity factor in concrete. My personal expertise is with mechanoluminescence, not concrete or stress/fracture of material so I will focus on the use of ML. Overall, this is an interesting application for ML and the research appears well carried out. Focusing on the ML I have one point that I would like addressed for better understanding/clarity.

What is time frame of crack propagation and what is the time resolution of the CMOD? The reader is informed that the video, that is capturing the ML, is 60 frames pes second, but how does the time resolution of the CMOD compare. Is it slower and if so would that account for any of the differences between the two signals in figure 4? Please explain. In addition, how long does the crack propagation take (i.e. how many data points do we have for each technique)? This will help the reader interpret the data better.

Author Response

In this paper, the authors explain the use of mechanoluminescence to use of ML technology to visualize and explain the fracture-process zones (FPZs) in concrete structures as well as calculate the stress intensity factor in concrete. My personal expertise is with mechanoluminescence, not concrete or stress/fracture of material so I will focus on the use of ML. Overall, this is an interesting application for ML and the research appears well carried out. Focusing on the ML I have one point that I would like addressed for better understanding/clarity.

What is time frame of crack propagation and what is the time resolution of the CMOD? The reader is informed that the video, that is capturing the ML, is 60 frames pes second, but how does the time resolution of the CMOD compare. Is it slower and if so would that account for any of the differences between the two signals in figure 4? Please explain. In addition, how long does the crack propagation take (i.e. how many data points do we have for each technique)? This will help the reader interpret the data better.

Response

Thank you for your thorough review and salient observations.

The CMOD was measured using TML Dynamic Strainmeters with sampling frequency of 5kHz. The CMOD data from TML Dynamic Strainmeters  was further synchronized with camera images and applied load using Multi-channel data link (MCDL) with 250 sample/sec. Therefore, each image was taken for every 0.004 seconds and therefore time in Figure 2 is 2.844 seconds. Since the concrete is brittle in nature the crack speed under 2000N/min loading rate is very high resulting in very short duration for crack propagation (around 0.064 seconds). As a result, out of 500 synchronized data points, around 16 images only depict crack propagation. Before synchronization, for 2.844 seconds, around 11,000 CMOD data can be acquired.

To address reviewer’s concern, text has been added to the revised manuscript: line, 164-170.

Reviewer 2 Report

This is an interesting article, it is dealing to mechanoluminescemce technology has the determine fracture, to visualize crack and to measure their lengths in structural material which makes it promising for monitoring the structural integrity of structures. However, a minor revision is needed before it can be considered for publication.

Some comments:

  1. I think in the introduction need to add the scientific works of PhD D.O. Olawale et al. (sensors system for concrete structures via mechanoluminescent technology).
  2. I suggest smoothing the curves in the figures, for example, Figures 2, 4, 8.
  3. In chemical formulas (subscript and superscript) need to be corrected in the literature lists, for example 3) ... thin-film SrAl2O4: Eu2 +, Dy3 +

Author Response

This is an interesting article, it is dealing to mechanoluminescemce technology has the determine fracture, to visualize crack and to measure their lengths in structural material which makes it promising for monitoring the structural integrity of structures. However, a minor revision is needed before it can be considered for publication.

Some comments:

  1. I think in the introduction need to add the scientific works of PhD D.O. Olawale et al. (sensors system for concrete structures via mechanoluminescent technology).
  2. I suggest smoothing the curves in the figures, for example, Figures 2, 4, 8.
  3. In chemical formulas (subscript and superscript) need to be corrected in the literature lists, for example 3) ... thin-film SrAl2O4: Eu2 +, Dy3 +

Response

Thank you for your thorough review and salient observations.

  1. The authors are very grateful for this valuable comment. As suggested, valuable work of PhD D.O. Olawale et al. (Development of a triboluminescence-based sensor system for concrete structures, Triboluminescent Sensors for Cement-Based Composites) has been cited in the revised version of the paper.
  2. The curves in Figure 2 have been smoothened. However, the time in Figure 4 and Figure 8 is a very short time (0.064 seconds), during which time there are 16 data images used in the graph. Thus it was not carried out because no clear characteristic equations for curve fitting are available at this stage. We hope the reviewer would understand this situation.
  3. The authors appreciate this comment and subscript and superscript in chemical formulas have been corrected throughout the manuscript.

Reviewer 3 Report

As an visible method, mechanoluminescence is used to monitor the fatigue crack of cement, which has practical significance. Kim and coworkers keep moving forward and make great progress in this field. The presented data is convincing and the anlysis seems reasonable. It suitable to be published in SENSOR. Minor considerations

1. What is the possible influence of thickness and uniformity of ML layers on results? and what is mode I stress ?

2. Details of mechanoluminescent materials SAOED, purchased or synthesized, main synthesis conditionsare recommended to provide.

3. Introduction part, in order to attract more attention useful references are recommended to be cited.
  (1)‘Triboluminescent Sensors for Cement-Based Composites’,   https://link.springer.xilesou.top/chapter/10.1007/978-3-319-38842-7_13
  (2) Chem Plus Chem, 2015, 80 (8), 1209-1215; Nano Energy, 2019, 55, 389-400;   Advanced Materials,2019, 31(7),1807062

Author Response

As an visible method, mechanoluminescence is used to monitor the fatigue crack of cement, which has practical significance. Kim and coworkers keep moving forward and make great progress in this field. The presented data is convincing and the anlysis seems reasonable. It suitable to be published in SENSOR. Minor considerations

  1. What is the possible influence of thickness and uniformity of ML layers on results? and what is mode I stress ?
  2. Details of mechanoluminescent materials SAOED, purchased or synthesized, main synthesis conditionsare recommended to provide.
  3. Introduction part, in order to attract more attention useful references are recommended to be cited.

(1) ‘Triboluminescent Sensors for Cement-Based Composites’,   https://link.springer.xilesou.top/chapter/10.1007/978-3-319-38842-7_13

(2) Chem Plus Chem, 2015, 80 (8), 1209-1215; Nano Energy, 2019, 55, 389-400;   Advanced Materials,2019, 31(7),1807062

Response

Thank you for your thorough review and salient observations.

  1. ML paint thickness should be very low comparative to the thickness of sample so that ML paint can act as a stress/strain sensing device, otherwise the larger thickness of paint could affect the deforming characteristics of structural materials.

        Uniform ML paint layers on the surface of structure is required to avoid              fluctuation in ML intensity. The thicker layer results in brighter ML pattern          as compared to thinner layer, which will ultimately affect while analyzing           stress/strain related phenomenon.

       The Mode I stress intensity factor is the most often used engineering                 design  parameter in fracture mechanics which reveal the fracture                     tolerance   of         materials used in bridges, buildings, aircraft etc.

       To address the reviewer concern, in the revised manuscript, lines 122-126         and lines  92-93 have been added.

  1. Since we have purchased commercially available SAOED, the name of company that synthesizes SAOED has been added to the revised manuscript, which is NEMOTO & CO., JAPAN.

        The information has been in lines 116-117.

  1. The authors are very grateful for this valuable comment. As suggested, the articles are cited in the revised version of the paper.